# A Gold Standard-Derived Modular Barcoding Approach to Cancer Transcriptomics

**DOI:** 10.3390/cancers16101886

**Published:** 2024-05-15

**Authors:** Yan Zhu, Mohamad Karim I. Koleilat, Jason Roszik, Man Kam Kwong, Zhonglin Wang, Dipen M. Maru, Scott Kopetz, Lawrence N. Kwong

**Affiliations:** 1Department of Translational Molecular Pathology, The University of Texas MD Anderson Cancer Center, Houston, TX 77030, USA; yzhu12@mdanderson.org (Y.Z.); mikoleilat@mdanderson.org (M.K.I.K.); 2Department of Melanoma Medical Oncology, The University of Texas MD Anderson Cancer Center, Houston, TX 77030, USA; jroszik@mdanderson.org; 3Department of Applied Mathematics, Hong Kong Polytechnic University, Hong Kong, China; mkkwong@math.uic.edu; 4Social Science Research Institute, Duke University, Durham, NC 27708, USA; zw277@duke.edu; 5Department of Anatomical Pathology, The University of Texas MD Anderson Cancer Center, Houston, TX 77030, USA; dmaru@mdanderson.org; 6Department of Gastrointestinal Medical Oncology, The University of Texas MD Anderson Cancer Center, Houston, TX 77030, USA; skopetz@mdanderson.org; 7Department of Genomic Medicine, The University of Texas MD Anderson Cancer Center, Houston, TX 77030, USA

**Keywords:** cancer, modules, barcoding, next-generation sequencing

## Abstract

**Simple Summary:**

Many resources exist to analyze cancer RNA data, but many of the algorithms and programs can appear as black boxes to non-bioinformaticians. To make RNA data more accessible, we here present modular barcoding, an approach predicated on the idea that cancer type-specific modules derived from high-quality, “gold standard” datasets will also be of high quality. Key to the use of these modules is their direct visualization, which can be done in spreadsheet programs in a color-coded way, essentially creating interactive heatmaps and visual gene set enrichments. We illustrate a variety of uses, including cancer subtype analyses, novel gene–gene and gene–clinical relationships, the inference of novel gene functions, and single-cell RNAseq analysis. Finally, we provide the tools for users to create their own modules, which will further improve their quality over time as single-cell RNAseq resolution advances. Modular barcoding is a user-friendly, tractable, yet powerful approach to make novel transcriptomic discoveries.

**Abstract:**

A challenge with studying cancer transcriptomes is in distilling the wealth of information down into manageable portions of information. In this resource, we develop an approach that creates and assembles cancer type-specific gene expression modules into flexible barcodes, allowing for adaptation to a wide variety of uses. Specifically, we propose that modules derived organically from high-quality gold standards such as The Cancer Genome Atlas (TCGA) can accurately capture and describe functionally related genes that are relevant to specific cancer types. We show that such modules can: (1) uncover novel gene relationships and nominate new functional memberships, (2) improve and speed up analysis of smaller or lower-resolution datasets, (3) re-create and expand known cancer subtyping schemes, (4) act as a “decoder” to bridge seemingly disparate established gene signatures, and (5) efficiently apply single-cell RNA sequencing information to other datasets. Moreover, such modules can be used in conjunction with native spreadsheet program commands to create a powerful and rapid approach to hypothesis generation and testing that is readily accessible to non-bioinformaticians. Finally, we provide tools for users to create and interpret their own modules. Overall, the flexible modular nature of the proposed barcoding provides a user-friendly approach to rapidly decoding transcriptome-wide data for research or, potentially, clinical uses.

## 1. Background

When analyzing bulk transcriptomic data, it can be a daunting task to deal with ~20,000 coding genes, or ~50,000 when including non-coding transcripts. Multiple efforts have been made to simplify analysis of the data, including the creation of modules that are either knowledge-based or derived unbiasedly from the data, as exemplified by the geneset collections c1–c8 in the MSigDb [1]; these include, for example, literature-based (c5), chromosome position-based (c1), perturbation-based (c2), curated immune (c7), and gene correlation-based genesets (c4). However, so far, such modules have primarily been used to plug into programs such as Gene Set Enrichment Analysis (GSEA) [2], Enrichr [3], or Cytoscape [4], and the average user may or may not have sufficiently intimate knowledge of each algorithm or geneset to know which results to reject or keep based on their biological and/or statistical quality.

In the following study, we suggest that correlation-based, cancer type-specific modules derived from high-quality, “gold-standard” datasets are a highly flexible and powerful means by which both bioinformaticians and non-bioinformaticians can rapidly interrogate transcriptomic data. Moreover, we expound on specific and novel uses of such modules to uncover previously unappreciated gene associations. Such modules can also be incorporated directly into spreadsheet-based programs, which in concert with color coding, statistics, and sorting, offer rapid visual outputs while displaying the actual gene values and names in each module. In this way, a non-bioinformatician can become deeply familiar with gene associations and, critically, rapidly generate and discard or keep hypotheses for downstream professional bioinformatics.

Such gold standard modules can be derived from any high-quality, high-quantity dataset such as The Cancer Genome Atlas (TCGA) [5] or International Cancer Genome Consortium (ICGC) [6], which feature uniform sample pipelines and stringent quality control standards for thousands of samples. In addition, modules derived from datasets such as xenograft or single-cell RNA sequencing data can further increase resolution at the tumor-intrinsic and stromal levels. Ultimately, modules from various sources can be intermixed and assembled into flexible barcodes which present multifaceted information for each sample. Here, we demonstrate that gold standard-derived modules can robustly serve a number of roles for the efficient and user-friendly interrogation of RNA-based datasets.

## 2. Methods

### 2.1. Datasets

We selected 19 TCGA cancer types for module creation. Overall, 14 of the cancers had >300 samples, which we selected as the cutoff for inclusion in pan-cancer analyses. Modules from all 19 cancers are shown in Appendix A. All RNAseq files, provided as TPM, were retrieved from the TCGA firehose portal (https://gdac.broadinstitute.org/ (accessed on 9 May 2024)), ending in “illuminahiseq_rnaseqv2-RSEM_genes_normalized”. Before module creation, all normal tissue samples were removed. A further 13 published and 1 unpublished datasets were used to demonstrate module utility, including 3 single-cell RNAseq datasets. The sources and download links are available in Appendix A.

### 2.2. Programs

We have generated several R and VB scripts to help facilitate in-spreadsheet analyses. All programs are available at https://github.com/lnkwong/Modular-Barcoding. Detailed descriptions of each program are included in the README, and brief descriptions are provided below.

### 2.3. Module Creation Program

The module_chooser.r script generates a large matrix of all gene–gene correlations for any dataset. In order to identify the strongest correlation clusters, for each gene, only the top 5% of genes with the highest correlation values to it are kept; all other genes for that gene are set to a correlation value of zero. This 5% cutoff prevents intermediately correlating genes from “pulling” genes out of their “proper” module. The resulting matrix is then clustered, and the tree is cut (default h = 0.8, can be varied), outputting gene module memberships as well as intermediate heatmaps. The reason for this overall approach is that straightforward, e.g., hierarchical clustering, K-means, or non-negative matrix factorization of genes on the expression data instead of the correlation values tends to produce very poor modules, especially for large datasets.

### 2.4. Median Centering and Heatmap Coloring Programs

The medcenter.r script allows a user to transform any dataset into a median-centered one, including separate columns giving the medians and averages for each gene. These median-centered values can then be colored in spreadsheet programs, for example, using the VB scripts we have provided in Heatmap Macro.xlsm, for either logged or unlogged data. This is very useful for pattern visualization, as it converts all values into fold-change data for direct comparison across genes (essentially an interactive version of R heatmap functions).

### 2.5. Module Correlation Program

The module_corr.r script performs 2 functions on data generated from the medcenter.r script (after removing the median and average columns). (1) It calculates collapsed single values for each module. We have previously published this method [7]; briefly, all genes in a module are correlated to a median module value iteratively, each time keeping only the top correlating genes (r > 0.6) to recalculate the median value until it reaches a fixed point; the final set of medians is taken as the collapsed values. (2) It correlates each gene to each collapsed module value. This generates a table that allows a user to manually examine whether some genes may correlate highly (positively or negatively) with multiple modules.

### 2.6. Additional Bioinformatics Analyses

GSEA was performed using the standalone GSEA program from https://www.gsea-msigdb.org/gsea/downloads.jsp (accessed on 9 May 2024) and run using default settings for preranked GSEA [2]. COCA was run with default settings using the coca R package (https://github.com/acabassi/coca (R version:4.3.1), accessed on 9 May 2024) [8]. Heatmaps were drawn with either standard R functions such as heatmap2 or using the Multiple Experiment Viewer (MeV) program (https://webmev.tm4.org/ (accessed on 9 May 2024)). All correlations throughout the manuscript are Spearman. All spreadsheet screenshots are from the program Microsoft Excel (Version:16.58).

## 3. Results

### 3.1. Establishment of Gold Standard-Derived Gene Modules for 14 Cancer Types

We systematically generated a distinct set of gene modules for each of 14 cancer types, using the high-quality and uniformly processed TCGA RNAseq datasets (Figure 1A). Within each cancer type, 20,501 genes were assigned to one module each; within each module, the genes are highly correlated to one another. Thus, when the genes are organized by module in a heatmap, they appear visually as demarcated bands of gene expression that highlight both the similar patterns within each module and the differences between modules (Figure 1B). Previously, we had interrogated MSigDb genesets to generate an annotated list of >1000 genes with high-confidence functions (e.g., CCNB1 as cell cycle, CD8 as immune, etc.). Manual comparison of this annotated list to our new modules in each cancer type readily revealed many with clear functional enrichments, such as cell cycle and immune signatures, which were then validated by GSEA. Moreover, the average correlation of each gene to its own module was high across modules—with the exception of genes with low absolute expression, consistent with our previous findings [7] (Appendix A). Thus, these modules create a high-level organization of genes that can reflect similarities of function within each module.

We first sought to identify conserved pan-cancer modules using a cluster of clusters [8] approach. To our surprise, only seven clusters of modules were identified as conserved across all cancer types (Figure 1C). Pathway analyses identified six of these as highly enriched for specific functions: immune, epigenetic, epithelial-to-mesenchymal transition (EMT), ribosome, mitochondrial/oxphos, and cell cycle. The seventh pan-cancer module (“unknown”) has not previously been described, and gene set enrichment analyses failed to uncover strong evidence of a dominant shared function, though one may yet exist. Overall, the finding of only seven conserved signatures is consistent with a high degree of cancer-type transcriptional specificity.

### 3.2. Cancer Module Relationships

Next, we sought to identify cancer type-specific pathways and their relationships to one another, to pan-cancer modules, and to the underlying biology. To do so, we began by correlating modules within a cancer to each other. To facilitate this, we collapsed each one down to a single “module score” [7]. Figure 2A,B shows representative examples, with a heatmap of modules clustered by correlation values matched to a heatmap of the collapsed expression data, arranged by standard subtyping schemes (CMS for COAD [9] and PAM50 for BRCA [10]). The approach reveals several associations. First, as expected, many modules correlate positively with one another, which is partly explained by the hierarchical tree-cutting step (Figure 1A) cutting some large modules into smaller ones, which are then “reunited” here. Interestingly, these small divisions tend to be preserved across cancer types, suggesting true differential patterns. Regardless, the correlation heatmap reveals not only families of related modules (such as those related to EMT), but also broader cancer type-specific correlations, such as anticorrelation between immune and oxphos in COAD.

Second, the relationships of modules to the underlying biology are easily visualized by their comparison to the subtype data. For example, COAD CMS4 is known to be enriched in EMT and immune signals [9], which is replicated by the EMT and immune modules here (Figure 2A); module #26 (“MSI-Low 2”) is low in the MSI-enriched CMS1, while a goblet cell module #123 is high in the goblet-enriched CMS3. Similarly, the BRCA basal subtype is known to be highly proliferative [10] and is reflected here by high expression of the cell cycle module #38 (Figure 2B), whereas the basal-high module #27 is enriched for known basal markers, including KRT5, KRT6A/B/C, and EGFR. The correlation heatmaps for the other 12 cancer types are shown in Appendix A. These associations further support the biological fidelity of the modules and lay a framework for their broader utility.

Third, we can assess how correlations between modules behave across cancers, by plotting the correlation values of the seven conserved pan-cancer modules across the 14 cancer types (Figure 2C, excluding “unknown”). As we previously reported [11], the epigenetic and oxphos modules consistent negatively correlate with one another; the reason for this is unknown, but it is preserved in many (but not all) non-TCGA datasets as well. We also found a high degree of consistent positive correlation between the ribosome and oxphos modules, which may reflect the demands of protein synthesis on ATP production, as well as between the immune and EMT modules, which may track with the degree of overall stromal presence. Overall, these analyses help clarify how modules are related to one another both within and across the 14 cancer types.

### 3.3. Application of Gold Standard-Derived Gene Modules to Other Datasets

Another advantage of the TCGA dataset is that all 14 cancer types exceed 300 samples, providing statistical robustness and accuracy for module membership. In contrast, most publicly available transcriptomic datasets involve fewer than 50 samples: it would be difficult or even impossible to generate high-quality modules directly from such small datasets, especially older microarray data. Thus, we sought to determine whether the gold standard-derived TCGA modules can improve the analysis of smaller transcriptomic datasets.

We started with two patient melanoma RNAseq studies of equal size: “JCI pre/on/post BRAF inhibitor” (n = 39) [12], and “JCOPO primary melanomas” (n = 39) [7] (Appendix A). For both datasets, we assigned each gene to its TCGA-derived SKCM module and first noted that the modules retained their general overall pattern as a heatmap (Figure 3A). Next, we found that modules with high intra-module correlation in the TCGA data tended to remain high in the smaller datasets and vice versa, supporting the hypothesis that the TCGA-derived modules capture generalizable gene-gene relationships (Figure 3B). Notably, the fresh frozen-derived JCI dataset showed significantly higher intra-module correlation values than the formalin-fixed paraffin-embedded (FFPE)-derived JCOPO dataset (Figure 3B), accurately reflecting data quality. Moreover, we could readily visualize known patterns in the data, such as increased immune module and decreased cell cycle module values in melanomas treated with BRAF inhibitor (“On BRAFi”) in the JCI dataset (Figure 3A), both of which rebounded after drug resistance; and increased cell cycle and decreased skin signatures in recurrent melanomas in the JCOPO dataset, reflecting their known higher mitotic rate and deeper Breslow thickness, respectively. 

Next, we applied the TCGA melanoma modules to two microarray-based (Affymetrix U133A 2.0, Santa Clara, CA, USA) datasets from frozen patient samples, GSE46517 (n = 120) [13] and GSE15605 (n = 74) [14] (Figure 3C). Unlike RNAseq, microarray data tend to suffer from non-uniform probe quality. To overcome this issue, we took advantage of the gold standard correlation information in each module to identify probes that correlated poorly in the microarray dataset. The best predictor of poor probe correlation within individual modules was the probe intensity (Figure 3D): plotting intensity versus correlation revealed a consistent optimum cutoff of ~50 (Figure 3E) for the platform. After removing low-intensity probes, the intra-module correlation compared to TCGA was roughly comparable to that of the FFPE RNAseq JCOPO dataset, reflecting the lower quality of microarray vs. RNAseq data (Figure 3F). Nevertheless, the TCGA-derived modules accurately reflected biology: in both datasets, higher expression of the skin module was clear in normal skin and primary melanomas compared to distant metastases, and, conversely, both primary and metastatic melanomas have higher cell cycle and melanocyte signature expression than normal skin (Figure 3C). Interestingly, both datasets also showed a previously unidentified correlation: a higher expression of the epigenetic module in metastases compared to primary melanomas. These observations suggest that TCGA-derived modules capture cancer type-specific biological fidelity and can highlight novel associations even in lower-quality datasets.

We also wish to demonstrate the user-friendliness of the modules, particularly for non-bioinformaticians. The module annotation can be readily aligned to expression data in a standard spreadsheet program, and the cells can be colored based on the gene expression values, in effect creating a “live” interactive heatmap; coloring can be done automatically through scripts (see Section 2). Figure 3G shows an example of how such a spreadsheet heatmap looks, using a 121-melanoma RNAseq dataset from Liu et al. from patients treated with anti-PD1 [15]. Illustratively, a single click of the “sort” function can instantly change disorganized expression data into a highly ordered, easily visualized heatmap (Figure 3G). When the samples are also sorted by overall survival, a clear visual correlation with the immune module emerges, which was validated by Kaplan–Meier analysis (Figure 3H) and consistent with the study’s findings [15]. Finally, the intra-module correlation of this dataset to the TCGA modules reaffirms high biological fidelity (Figure 3I). 

We have applied our modules to over 100 public and proprietary datasets of various cancer types, of which the above are representative of the high degree of correlation, data organization, and capacity to rapidly derive known and novel biological insights. We have provided the module memberships for all coding genes across 14 TCGA cancer types in Appendix A. Below, we illustrate several different uses and advantages of gold standard-based modules.

**Figure 3 cancers-16-01886-f003:**
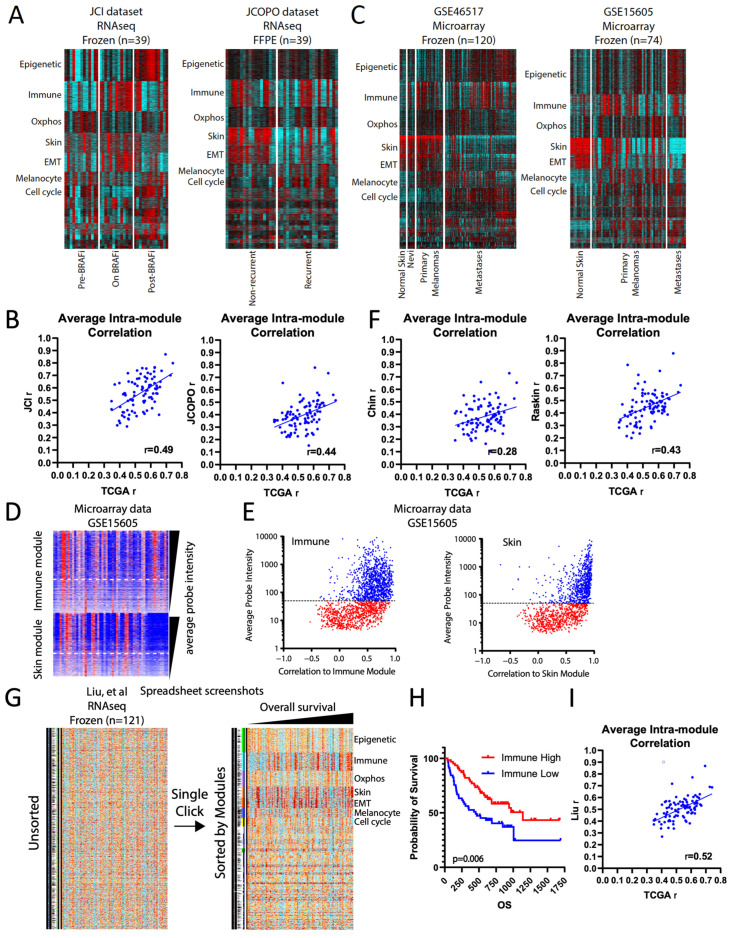
TCGA-derived modules enhance the analysis of smaller datasets. (**A**) Heatmap of TCGA-derived modules for the JCI and JCOPO melanoma datasets. (**B**) Average intra-module correlation values (i.e., each gene within a module correlated to its collapsed module value), comparing the JCI and JCOPO datasets to the original TCGA values. (**C**) Heatmap of TCGA-derived modules for the GSE46517 and GSE15605 datasets. (**D**) Heatmap of genes in the immune and skin modules from GSE15605, arranged by decreasing average probe intensity within each module. (**E**) Scatterplot of average probe intensity versus the intra-module correlation values. Lower correlation values imply poorer probe quality. Blue = high quality probes, red = poor quality probes. (**F**) Average intra-module correlation values, comparing the GSE46517 and GSE15605 datasets to the original TCGA values. (**G**) Screenshots from spreadsheet layouts of the data from Liu et al. with cells colored from high (red) to low (blue) log2 fold-change value. One-click sorting of the data by module assignments reorganizes the data. (**H**) Kaplan–Meier survival plot comparing samples with high versus low collapsed immune module values, using the median value as the cutoff, from Liu et al. (**I**) Average intra-module correlation values, comparing the Liu et al. dataset to the original TCGA values [15].

### 3.4. Modular Barcoding Recreates and Extends Known Cancer Subtypes 

We reasoned that rationally selected modules could capture established cancer subtyping schemes. First, we asked whether our colon cancer modules could recreate the four COAD CMS subtypes [9]. We identified a subset of six modules that, using non-negative matrix factorization (NMF), could assign samples to their CMS subtype with a concordance of 86%, using only the collapsed scores (Appendix A). Conversely, arrangement of the six modules by their previously defined CMS subtypes also readily demonstrates their fidelity (Figure 4A). For example, we once again observed known CMS enrichments for the collapsed immune, EMT, and goblet modules. We next asked whether module barcoding could also identify novel associations. For example, modules #84 and #54 show high expression in CMS1 and moderate expression in CMS3; both modules display an enrichment for canonical ERK signature genes, including DUSP4, ETV5, FOSL1, PHLDA1, and PLAUR [16,17], consistent with the known high enrichment for BRAF mutations in CMS1 and KRAS mutations in CMS3. To our knowledge, this association has not previously been reported.

Similar module-led subtyping of breast cancer (PAM50 subtypes) [10], glioblastoma (intrinsic subtypes) [18], and ovarian (TCGA subtypes) [19] cancers yielded similar sets of modules that readily matched established subtyping schemes. Several components become immediately clear from known module functions, such as cell cycle in the PAM50 classification, a well-established component of the signature. In GBM, the established mesenchymal subtype is seen to be composed of both a mesenchymal (#54) and an immune module (#32); despite their high degree of overlap, these two modules nevertheless show a clear difference in pattern. A more novel inference is that ovarian module #136 is enriched for basal markers, potentially suggesting module #130 as luminal-like.

We next asked whether our modules could help create subtyping schemes, focusing on melanoma. In principle, subtyping assumes a certain degree of mutual exclusivity of signatures, such as the well-established one between EMT and melanocyte signatures in melanoma [20,21]. However, despite recreating this mutual exclusivity with our modules, we found a sizable portion of samples that expressed neither EMT nor melanocyte genes; examination of other modules identified #35 as high in the majority of these doubly low samples and mutually exclusive with the melanocyte module (Figure 4A). Module #35 contains multiple genes well established to be downstream of MAPK signaling, including ETV1, DUSP6, SPRY2, and SPRY4 [16,17]. Consistent with this, in three melanoma datasets examined, module #35 was higher in RAS/RAF/NF1-mutant than triple-wild type samples (Appendix A). If we then use these three modules (EMT, melanocyte, ERK) to generate five subtypes, we can readily recreate this barcode scheme in independent datasets (Figure 4B). Strikingly, the MEL-HI subgroup consistently showed poor survival, while the “Low” subgroup showed high survival across all three datasets (Figure 4C. The three ERK-HI/EMT-HI groups were in between and were consolidated for clarity and because the smaller datasets had too few samples. We therefore propose this as one possible subtyping scheme for melanoma. Moreover, adding more modules identified the immune module as enriched in EMT-HI samples and the oxphos module as enriched in MEL-HI samples (Figure 4B). 

Overall, the modules in each subtyping scheme can be read as a modular barcode describing the sample’s relative status, for example: high cell cycle, low EMT, medium immune, and so on, but this barcode is not static. For example, non-coding RNAs can also generate modules, which tend to mirror mRNA modules (Appendix A). Much like the SKCM example (Figure 4B), any combination of modules can be used to tell a broader story about individual samples; this could provide concise information to initiate research or perhaps even clinical questions for specific patients. 

### 3.5. Candidate Novel Members of Pan-Cancer Modules

We next took advantage of the seven identified pan-cancer modules to nominate novel pathway members, based on the conservation of gene membership across cancer types. For example, the cell cycle module contains a minimum of 252 core genes (Appendix A). Intersection of this core gene list with knowledge-based cell cycle pathway lists and literature searches identified 11 genes with no previously described cell cycle function, as of our initial analysis in mid 2020 (Figure 5A). During our subsequent analyses, 3 of the 11 genes were validated as having cell cycle or DNA damage repair (DDR) functions, as DDR is strongly coupled to the cell cycle: FAM72A as a novel DDR protein involved in antibody diversification [22,23]; the FAM111B protease as a regulator of cell cycle via p16 degradation [24]; and C17orf53 as a component of inter-strand crosslink repair [25]. Additionally, knockdown of C5orf34 or TMEM194A have been shown to affect the cell cycle [26,27]. These studies collectively support our 11 cell cycle candidates.

For example, we looked closer at C16orf59, also known as TEDC2. In the BioGRID database [28], the C16orf59 protein product physically interacts with 24 proteins, 2 of which, KNSTRN and MIS18A, are also core members of the cell cycle module (Figure 5B). KNSTRN binds and helps properly align and segregate kinetochores during mitosis [29,30], while MIS18A also acts at kinetochores during mitosis by recruiting CENPA [31]. Moreover, C16orf59 has been described as pro-oncogeneic [32] and also interacts with TUBE1, which maintains centriole microtubule integrity [33]. These data suggest that deeper functional analysis of C16orf59 is likely to yield new mechanistic insight into the mitotic machinery.

Second, the epigenetic module has, to our knowledge, not been described previously. We identified 579 core members of this module, containing a strong enrichment for chromatin modifying factors including many which are recurrently mutated in cancer, such as ARID1A, ARID2, PBRM1, SUZ12, and TET2 [34] (Figure 5C), as well as other well-established components of the SWI/SNF, NuRD, TFII, and TFIII complexes, and more. Surprisingly, key members of the oncogenic PI3K and MAPK pathways—PIK3CA, PDPK1, MAPK1 (ERK2), PTPN11 (SHP2), SOS1, NF1, and MAPK8 (JNK1)—are also core members of the epigenetic module, suggesting a possible link between their transcriptional regulation and chromatin remodeler expression.

In total, pan-cancer modules allow us to not only nominate novel functional members but also identify unexpected memberships. The full list of core members of all seven of the pan-cancer modules is listed in Appendix A. 

### 3.6. Gold Standard-Derived Modules Simplify and Speed Up Hypothesis Generation and Testing 

Next, we wish to expand on the utility of spreadsheet program-based analyses (Figure 3G) as an efficient hypothesis generation and testing tool. This approach primarily utilizes the following four features in concert: (1) gold standard-derived modules, (2) coloring of cells in the spreadsheet to create an interactive heatmap, (3) the spreadsheet program’s built-in statistical functions (e.g., correlation, average, median, standard deviation, t-test, etc.), and (4) the ‘sort’ function. This allows for directed visual pattern-seeking that can rapidly give rise to new hypotheses, followed by statistical testing to efficiently discard or keep those hypotheses for deeper downstream bioinformatic analyses. Below, we give several examples of this process.

First, we revisited the 121 frozen RNAseq melanoma dataset [15] (Figure 3G). If we sort the samples by overall survival (OS) and the genes by their Spearman correlation to the OS, and since the module membership is color-coded for known modules, it can be rapidly seen that light blue immune module genes are enriched at the top of the list (Figure 6A). At the bottom of the list, the strong green presence indicates an enrichment for a substantial subset of the epigenetic module, generating a novel association in this dataset with worse survival. In effect, this spreadsheet-driven approach is able to function like a basic Gene Set Enrichment Analysis (GSEA), which also relies on identifying enriched geneset membership along a ranked axis. Since any modules can be applied to any sorting of statistics (in this case the Spearman rho, calculated using native spreadsheet functions), the modules provide a way to rapidly conduct basic but effective GSEA simply by creating statistics and using the spreadsheet’s sort function. To confirm this, we inserted both our melanoma-specific and pan-cancer modules into the .gmt files of the MSigDb geneset collections c2 and c5, allowing for direct comparisons. Both the melanoma-specific and pan-cancer immune modules notably outperformed existing immune genesets (Figure 6B), and the melanoma-specific epigenetic geneset was also highly ranked, confirming the color-based observation in Figure 6A. Moreover, our ribosome modules scored highly as negatively correlated with overall survival, to our knowledge a novel observation. The actual spreadsheet used to create Figure 6A is provided in Appendix A as an example.

Second, we demonstrate that additional data such as mutations can easily be added and visualized. In the OEP001105 cholangiocarcinoma (CCA) RNAseq dataset of 256 frozen samples [35], we added and color-coded the mutation status of driver genes. We then refined and applied a previously published CCA Subtype signature, sorted all samples by that signature, and sorted all genes by their correlation to that signature (Figure 6C). The sorting of samples immediately visually confirmed the known association of the CCA Subtype 1 signature with FGFR2, IDH1/2, ARID1A, and BAP1 mutations, and negative correlation with KRAS and TP53 mutations [36]; this has not previously been demonstrated for this particular dataset. Moreover, our CCA1 signature module outperformed two existing CCA1 signature genesets in MSigDb c2 (Oishi and Andersen) in GSEA and identified novel correlations with our epigenetic and oxphos (positively correlated) and basal (negatively correlated) modules (Figure 6D). By perusing the sorted gene expression list, we found that the immunosuppressive gene VTCN1 (encoding B7H4) was also strongly correlated with the CCA1 signature (r = 0.7). We have previously reported high B7H4 protein levels in BAP1-mutant CCA [37]; thus, through only visual inspection of the RNAseq spreadsheet, we happened organically upon an orthogonal validation of this association (Appendix A).

Third, we show an example of the flexibility of module usage in an in vitro drug treatment setting. We combined the data from three independent studies on KRAS G12C inhibition (G12Ci) in six non-small cell lung carcinoma (NSCLC) cell lines [38,39,40], normalizing within each cell line to DMSO. One way to rank G12Ci-regulated genes across is by fold-change, but this can be confounded by large differences in fold-change effects between cell lines and/or low numbers of replicates. Instead, we ranked all genes within each sample and took the average rank across samples, providing a non-parametric ranking (Figure 6E). Cell cycle module genes (yellow) were enriched in genes downregulated by G12Ci as expected, as well as an Erk gene module derived from an independent dataset [16]. Next, we divided the values of cell lines with acquired resistance by those of their parental sensitive cells—both treated with G12Ci—creating a column of fold-change differences. If we now re-sort by their average ranks, they identify modules associated with the resistance phenotype: the Erk and cell cycle modules are again enriched, indicating their expected reversion, but they also find the EMT module #26 enriched as well (Figure 6F), including the well-established MAPK inhibitor resistance marker AXL [21,41]. Thus, this analysis rapidly identified EMT as a top candidate resistance mechanism in these cells that was not reported in the associated study [38]. Moreover, as the resistant cells are highly enriched for cell cycle changes, we took the opportunity to show (Appendix A) that our TCGA-derived pan-cancer cell cycle module exhibits a much higher enrichment score in GSEA and tighter visual enrichment when directly compared to existing cell cycle-related genesets in MSigDb c2 and c5. Overall, this combined dataset demonstrates the flexibility, quality, and speed by which different patterns can be found and relationships within and between samples can be queried using native spreadsheet functions plus gold standard-derived modules.

Finally, we demonstrate the benefit of using multiple module sets in one analysis. In examining the TCGA bladder cancer (BLCA) dataset [42], applying all 14 cancer type module sets identified a clear group of modules converging on a basal signature in 7 additional cancer types, including prostate (PRAD) and breast (BRCA), which, together with BLCA, are commonly divided into basal and luminal subtypes [43] (Figure 6H). Using these eight cancer types, we defined a set of 76 core basal genes (Appendix A), which include the classic markers KRT5, KRT6A/B/C, and DSG3. By contrast, no core genes could be defined as a cross-cancer luminal module, suggesting that unlike the basal state, the luminal state is highly cancer type-specific. Next, we used GSEA to compare our “pan-cancer” basal module to all existing basal cancer signatures in MSigDb c2 (all derived from breast cancer data) across four TCGA cancer types. Our basal module scored the most consistently across cancer types (Figure 6H), suggesting that it is indeed more applicably pan-cancer than the breast cancer-derived basal genesets. These results stand in contrast to the reverse approach of starting with a presumptive basal/luminal signature and enforcing membership across cancer types, which results in both a signature and sample calls that are confounded by non-basal genes such as cell cycle constituents [43]. Instead, our approach organically defined a clean basal signature and suggests that it is only a dominant signature in a subset of cancers, explaining its absence from the pan-cancer module list (Figure 1C). 

We emphasize that such observations can come from a single click of the ‘sort’ button on statistical data that also instantly sorts the correlated heatmap containing both colors and the actual numerical values. This, in turn, enables rapid additional statistical calculations, building directly on the results of the previous, and so on iteratively. In this way, hypotheses can be rapidly generated, tested, and kept or discarded.

### 3.7. Modules as Decoders between Signatures

We hypothesized that modules could also act as “decoders” between distinct gene signatures by providing a common scaffold for comparisons. We applied this to colon cancer, comparing two published subtyping regimens: the CMS subtypes [9] derived from bulk tumor transcriptomes, and the CRIS subtypes [44] derived from patient-derived xenograft (PDX) data in which only human tumor RNA is captured and mouse stromal RNA is omitted. The CMS and CRIS subtypes are largely discordant with each other, compounded by a lack of microsatellite instability (MSI) samples—which are enriched in CMS1—in the PDX dataset from which CRIS was derived. 

In order to understand the relationship between the two subtyping schemes, we compared their established gene signatures (n = 564 genes in CRIS; n = 500 genes in CMS). Few overlaps in gene members were observed, making it difficult to draw direct connections. However, TCGA-derived modules provide a comparative scaffold: shared modules can connect genes that otherwise are not obviously related (Figure 7A). Using this approach, we discovered that both subtyping signatures contained members of modules that are highly specific to MSI (i.e., modules #47, #26, and #84, Figure 7A), despite the lack of MSI samples in the CRIS-originating data. This implies that the MSI-specific modules represent a true biological signature, rather than an artifact of CMS1 samples being a different type of cancer. To validate this, we re-generated TCGA modules omitting CMS1 samples, and still identified a module, #301, that strongly overlaps with MSI-associated module #47 (Appendix A, “COAD-CMS1”).

Further analysis identified CMS and CRIS gene subsets both connected to the TCGA immune module. Given the lack of immune cells profiled in the PDX data (since they are mouse-derived), we asked whether these genes may instead represent a tumor-intrinsic immune signature. We therefore generated new modules from the PDX data and identified one module, #64, as having a significant overlap in gene members with the TCGA immune module (Figure 7B). Analysis of the 53 overlapping genes identified an enrichment for proteins with tumor-intrinsic immune functions such as B2M, MHC class I proteins, and interferon family members. To validate this observation further, we made use of a tumor-intrinsic PDX RNAseq dataset that includes MSI samples (manuscript in preparation, D.M., S.K.). Assessment of the 53 tumor-intrinsic immune genes confirmed an overall upregulation in MSI samples (Figure 7C), suggesting that MSI colorectal cancers may have an intrinsic immunogenicity.

A recent study also identified the iCMS tumor cell-intrinsic subtypes from single-cell sequencing data [45], which we re-analyzed in reference to our modules. First, we found that our tumor-intrinsic 53 immune gene list was indeed expressed more highly in MSI-like iCMS3 cells than in microsatellite stable (MSS)-like iCMS2 cells (Figure 7D). Second, we observed that our goblet module #123 and ERK module #84 genes were almost entirely higher in iCMS3—consistent with their high levels of KRAS/BRAF mutations—while our MSI-low modules #26 and #47 (Figure 4A) genes were almost entirely higher in iCMS2 (Figure 7D), suggesting that these four modules accurately captured tumor-intrinsic signatures. The iCMS report suggested that iCMS3 signature genes may represent metaplastic transdifferentiation towards a stomach-like lineage [45,46], but examination of the genes instead suggested specificity to normal colon goblet cells, including TFF1, TFF2, S100P, and SDR16C5 [47]. Indeed, our module #123 is highly enriched for normal goblet cell markers [47] (Figure 7E), including the master goblet regulatory transcription factor ATOH1 [48]. Overall, these examples illustrate the ability of gold standard-derived modules to act as a “decoder” to reconcile discordant gene signatures and subtyping schemes. 

### 3.8. Modules from Single Cell RNAseq

Our gold-standard module approach also works on single-cell RNAseq (scRNAseq) data. As an example, we have generated modules from a single-cell RNAseq of normal embryonic mouse liver [49], GSE90047, whose time course dataset provides a spectrum of liver cells from bipotential hepatoblasts to its two differentiated cell fates, hepatocytes and cholangiocytes. Non-liver cells were excluded. As expected, while some genes are strictly expressed in either hepatocytes or cholangiocytes, others are shared with hepatoblasts or in both differentiated cell types (Figure 8A). Assessment of the hepatoblast module revealed a strong enrichment for ribosome module genes (Figure 8B) as well as a strong correlation to the cell cycle module, consistent with the known biology (Figure 8A). This also illustrates that although the cell cycle and ribosomal modules are highly correlated, they remain largely distinct (Figure 8B). 

We then asked whether these normal mouse-derived scRNAseq modules can be informative for a bulk human cancer RNAseq dataset, using the combined cholangiocarcinoma (CHOL) + hepatocellular carcinoma (LIHC) TCGA data [11,50] as an example. The modular approach demonstrates a high proportion of genes expressed in normal mouse hepatocytes as being retained as higher in LIHC and bulk normal human liver, and normal mouse cholangiocyte genes as being retained as higher in CHOL (Figure 8C,D), in spite of the transcriptional dysregulation inherent in cancer and the cross-species differences. Interestingly, the hepatoblast module did not retain a cohesive structure in CHOL/LIHC (aside from ribosome genes), suggesting that these cancers do not, as a general rule, experience dedifferentiation back into a hepatoblast-like state. Overall, our results demonstrate that the modular approach can help simplify even large scRNA datasets for practical applications.

## 4. Discussion

### 4.1. Gold Standard-Derived Modular Barcodes Are Tractable Multi-Use Tools

In this study, we describe the creation, multi-utility, and flexibility of a modular barcoding system based on gold standard-derived, cancer type-specific modules. In previous reports where similar correlation-based modules have been proposed [51,52,53], the visual patterning of the modules and the identity of their gene memberships are largely hidden within summary figures. Here, we sought to make our modules as accessible as possible, not only in supplemental tables, but also in the form of expository heatmaps and visually direct associations with individual genes, clinical data, perturbation phenotypes, mutations, and with other modules. Moreover, we sought to illustrate multiple and varied uses of modules that can readily be applied by non-bioinformaticians, bolstered by our demonstration that our modules consistently outperform existing genetsets. Users can also generate their own modules from any sufficiently powered dataset. 

There are, however, several important considerations for our approach. First, it is often worth generating multiple different sets of modules from the same dataset, depending on the biological question. An example given above is removing the CMS1 samples from COAD, which revealed that apparently CMS1-specific modules were not actually dependent on CMS1 inclusion, suggesting that they capture a broadly relevant COAD signature. As another example, modules generated from the GTeX normal tissue gene expression database [54] tend to be skewed towards genes expressed in the brain due to the overrepresentation of brain sections; this has the unintended effect of obscuring modules that would otherwise be more specific to other tissue types. Balancing the number of brain samples included allows for the generation of module sets that reflect broader tissue-specific patterns. 

Second, our modules assign only one gene to one module, which may artificially “force” genes into modules they do not belong in or underrepresent genes that should belong to more than one module. To minimize these issues, we have developed an R program, “module_corr”, that outputs correlations of each gene to each module and aligns them to the heatmap; in this way, individual genes can be intuitively interrogated for their associations to multiple modules, implying, in some cases, multiple memberships. An example module_corr output for BLCA is shown in Appendix A, and a link to the program is provided in the Section 2. 

Finally, we emphasize here that it is not the goal of each module to represent “biological truths”, i.e., that they comprehensively capture a particular pathway, function, or cell type. Rather, they provide visual and statistical cues that might lead to biological truths. To that end, we have provided a link to the R program “module_chooser” in the Methods section for creating de novo modules. In such a way, end users can harness the flexibility of the system to suit module selections to their specific biological questions. 

### 4.2. Modules and Spreadsheet Analyses Speed Up the Hypothesis-Seeking Phase of Projects

In a typical collaboration, a biologist first asks a specific question, and the bioinformatician runs the query and provides a result, sometimes after a few hours, sometimes after a few days. The biologist examines the result and generates a second question, and so on back and forth over the course of days or weeks. During this time, the algorithm used is often no more than a black box to the biologist, and they must either probe deeper into its workings and quality controls, or else trust the results implicitly. 

In the scheme that we are advocating here, a module-enabled, spreadsheet-based approach can drastically shorten the question-answer response times—with, ideally, the biologist acting as their own bioinformatician during this hypothesis-seeking phase. The initial question gets posed and, through statistical functions in the spreadsheet and sorting the results, then viewing the color and statistical patterns in the fluid heatmap and modules, dozens of cycles of questions and answers can be run through in less than an hour, rather than over weeks. Moreover, the familiarity with the non-black box approach builds confidence and allows for corrections and quality control on the fly. Now, it is true that the answers provided in each cycle by this approach are relatively crude and underpowered compared to more powerful algorithms, but at this phase, this is more than sufficient—generally, a yes or no answer, or a suggestive statistical or visual pattern, will suffice to least tentatively keep or discard a given hypothesis. The best hypotheses are then finally posed to the bioinformatician to provide more in-depth and accurate results.

Stated another way: the biologist seeks the best questions and the bioinformatician provides the best answers. Moreover, in much the same way as a sculptor learns about the clay they are working with through direct touch with their hands, so too can a biologist working directly with the interactive transcriptomic data itself help shape the outcome of the analysis, by gaining an intimate familiarity with the genes and their qualitative and quantitative, context-dependent relationships.

### 4.3. Modules Can Assist with Quality Control

A large number of powerful algorithms exist to interrogate transcriptional data, yet the outputs are often static images, summary tables of information and statistics, or lists of genes and *p*-values, often leaving a non-bioinformatician poorly equipped to meaningfully quality control (QC) the results. For example, virtually any dataset can output top lists of differentially expressed genes (DEGs) with strong *p*-values, but that does not mean that these DEGs are trustworthy or biologically relevant; indeed, visualizations such as volcano plots can look virtually identical between high- and poor-quality results. Such problems can arise from either quality issues with the dataset itself (such as poor sample quality, the presence of technical artifacts, or even improper experimental design such as leaving out critical controls) or with the analyses.

For an example of the latter, in GSEA, it can be difficult to know which MSigDb genesets are informative and which may be misleading. For instance, enrichment for a geneset with a name such as “viral lifecycle” might appear to suggest a role for viral processes, but the actual genes reveal that it is simply a cell cycle geneset. As another example, genesets such as the “PI3K” or “MAPK” pathway often include genes whose primary activity readout is post-translational rather than transcriptional, such as AKT1/2/3, KRAS, or RAF1, yet are nonetheless applied to transcriptional data. Furthermore, geneset enrichment is, to some degree, predicated on the assumption that functionally related genes within a geneset correlate with one another or are at least coordinately regulated, but that is untrue of many genesets. As a final illustration, the MSigDb Hallmarks collection has only 50 genesets, which means it frequently misses out on obvious pathway enrichments, yet many non-bioinformaticians take the Hallmarks results at face value.

We propose that gold standard-derived modules can help solve or improve many of these issues. First, regarding data quality, sorting the data and viewing modules can quickly help QC a dataset: (i) if genes do not correlate well within modules that are expected to be conserved (e.g., immune in in vivo datasets); (ii) if certain modules are expected to be enriched, but are not; (iii) if specific samples show strange patterning across modules; or (iv) if the patterning highlights, say, missing controls or widely differing replicates, then these could possibly indicate dataset QC issues worth pursuing. Second, regarding geneset quality, gold standard-derived modules are already based on transcriptional correlation values, which means that they are particularly amenable to module enrichment algorithms which presuppose coordinate gene regulation; indeed, these modules can be directly added to .gmt files and perform well in GSEA. Finally, regarding the quality of bioinformatics results (e.g., DEG lists, geneset enrichments, clustering, etc.), we suggest that porting them back into the spreadsheet framework provides a sufficiently accurate “eyeball test” of color pattern visualizations and their associated statistical outputs to rapidly decide whether or not improvements are needed to the analysis design.

## 5. Conclusions

In summary, even large-scale transcriptomic data can be rendered accessible through organization via biologically relevant labels such as gold standard-derived modules. We emphasize that such modules are not limited to the TCGA dataset, and that the examples shown here of pan-tissue modules, novel gene memberships, barcodes, and decoding can be broadly applied within and beyond cancer, including to other disease states, normal tissues, and cross-species analyses. Overall, we pose modular barcoding as an approach by which to streamline, accelerate, and make accessible transcriptomic data for more accurate and efficiently collaborative analyses.

## Figures and Tables

**Figure 1 cancers-16-01886-f001:**
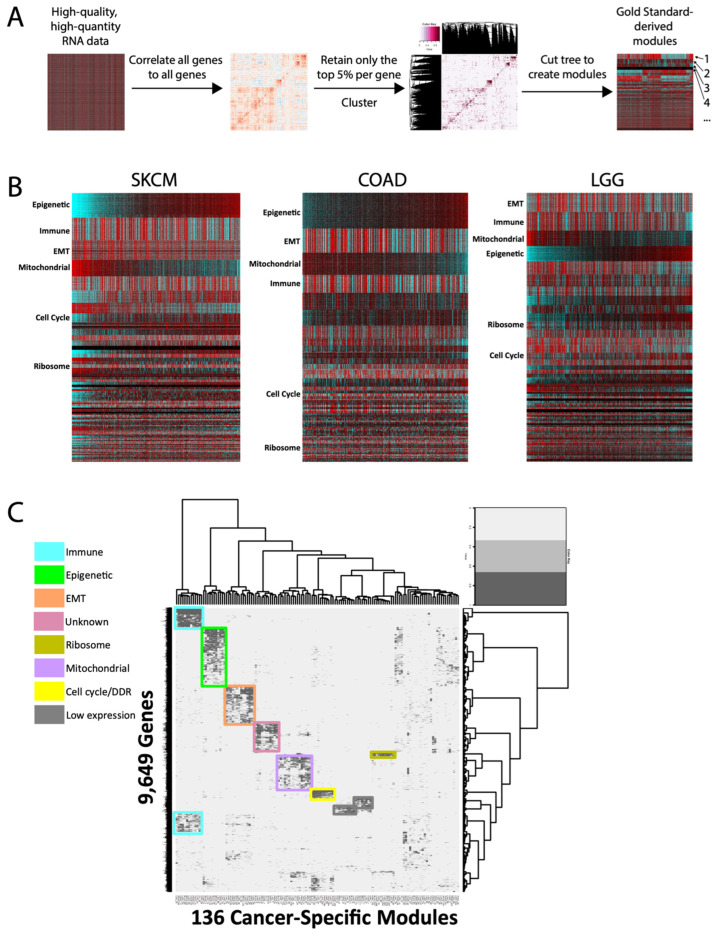
Module generation and cross-cancer conservation. (**A**) Overall schematic of module generation. Numbers represent hypothetical modules. (**B**) Representative heatmaps of modules from 3 TCGA cancer types. Columns are samples, and rows are genes. Modules are arranged in decreasing size and arbitrarily sorted the by epigenetic module. SKCM = melanoma, COAD = colorectal, LGG = low-grade glioma. (**C**) Cluster-of-clusters analysis for 136 cancer-specific modules across 11 cancer types, for 9649 genes. Overall, 7 conserved pan-cancer module clusters are highlighted, as well as 2 modules whose only commonality is low expression across cancers.

**Figure 2 cancers-16-01886-f002:**
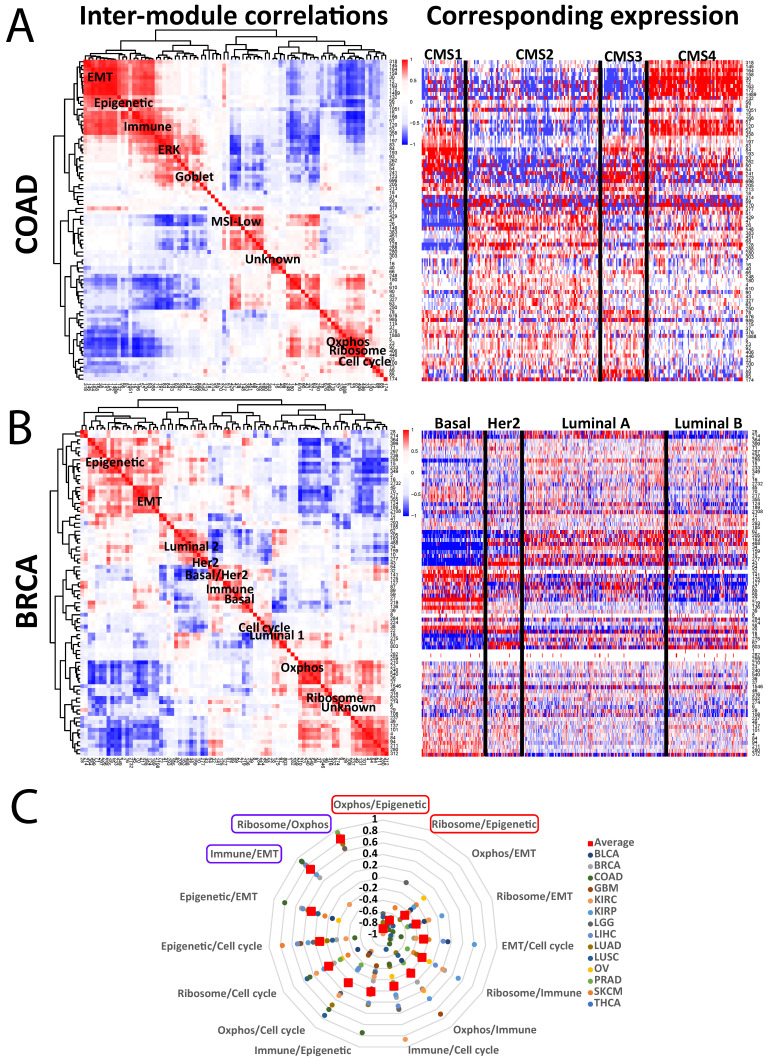
Cancer-specific associations between modules. (**A,B**) Left: Representative inter-module correlation heatmaps from (**A**) COAD and (**B**) BRCA (breast cancer). Each square represents the Spearman correlation value between 2 modules. Only modules >24 genes in size are included. Right: the corresponding expression heatmap for each module, organized by established subtyping schemes for each cancer. Specific modules are highlighted, including the 7 pan-cancer modules. (**C**) Correlation values for each of 6 pan-cancer modules to each other, for each TCGA cancer type.

**Figure 4 cancers-16-01886-f004:**
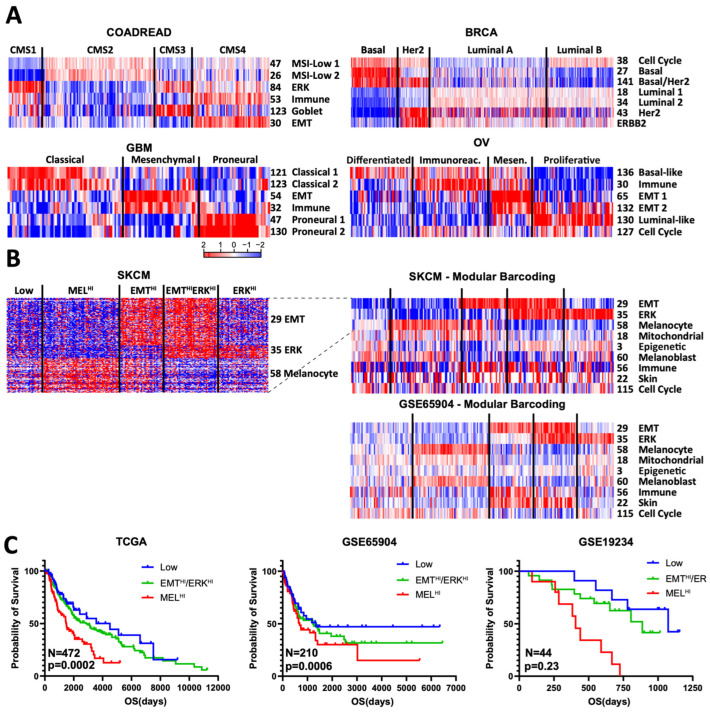
Modular barcoding recreates and extends cancer subtyping schemes. (**A**) Minimal selected modules used to recreate known subtyping schemes: CMS for COAD, PAM50 for BRCA, classical/mesenchymal/proneural for GBM (glioblastoma), and TCGA for OV (ovarian). (**B**) Proposed module-based subtyping scheme for SKCM. Left: heatmap of all genes in the 3 proposed modules; right: collapsed modules with extended barcoding using TCGA and GSE65904 melanoma datasets. (**C**) Kaplan–Meier survival plot for 3 datasets based on the proposed subtyping scheme. EMT^Hi^/ERK^HI^ subtypes have intermediate values and are combined for clarity and due to small sample numbers in the smaller datasets.

**Figure 5 cancers-16-01886-f005:**
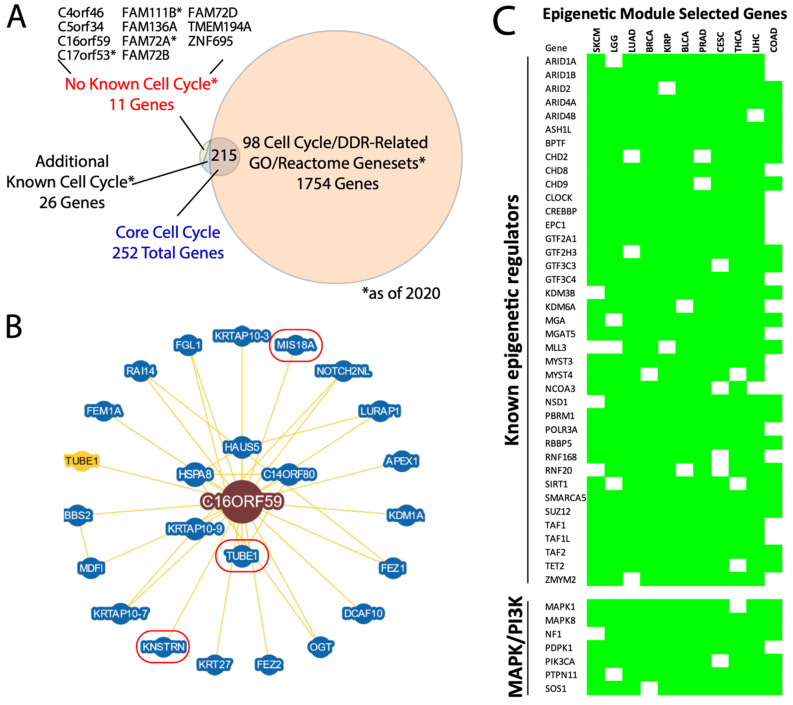
Pan-cancer modules nominate novel gene memberships. (**A**) Venn diagram of the 252 core cell cycle genes compared to knowledge/literature-based assessments of their known roles in the cell cycle machinery. MSigDb cell cycle/DDR-related genesets provided one layer of assessment and were independently validated in the literature. Overall, 11 genes as of 2020 had no demonstrated cell cycle role, of which 3 with asterisks were later shown to have such roles. * = as of 2020. (**B**) BioGRID protein interaction diagram for C16orf59. Three proteins with known cell cycle roles are encircled in red. (**C**) Selected core epigenetic module genes, highlighting both known chromatin-modifying/-interacting genes and known MAPK/PI3K pathway genes. The green cells indicate membership of that gene in the epigenetic module for each of the 11 cancer types listed.

**Figure 6 cancers-16-01886-f006:**
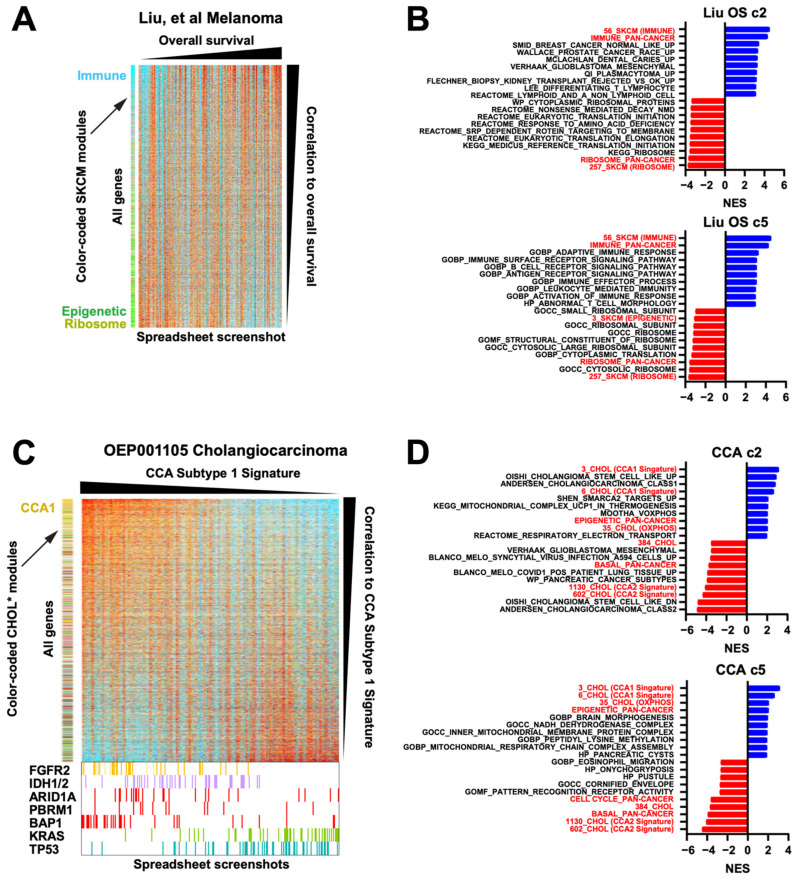
Module-facilitated spreadsheet analyses of RNA data enable rapid hypothesis generation and testing. (**A**) Spreadsheet screenshot of the Liu et al. melanoma dataset where the columns (samples) have been sorted by overall survival and the rows (genes) have been sorted by the Spearman correlation value. SKCM modules along the left are color coded [15]. (**B**) GSEA top results from MSigDb c2 and c5 geneset collections with our modules inserted into the .gmt, for correlation to OS from the Liu dataset, sorted by NES. All shown genesets have a pval = 0. Red = our modules. (**C**) Spreadsheet screenshots of the OEP001105 cholangiocarcinoma dataset, where the samples are sorted by their CCA Subtype 1 signature and genes are sorted by their Spearman correlation to that signature. CHOL modules (* manuscript in preparation, L.K.) along the left are color coded. At the bottom, enlarged from the same spreadsheet, are the mutation status of known driver genes. (**D**) GSEA top results for correlation to the CCA1 signature from the OEP cholangiocarcinoma dataset, sorted by NES. All shown genesets have a pval = 0. Red = our modules. (**E**) Spreadsheet screenshot of combined data from 3 NSCLC G12Ci datasets, normalized to DMSO within each cell line. Genes are sorted by the average rank of each gene’s fold-change value on G12Ci vs. DMSO. KRAS knockdown, MEK inhibitor, and G12Ci-resistant cell line data are included. Two module sets, one LUAD-specific and one ERK-specific, are along the left side. (**F**) The same dataset showing only the top 200 genes when sorted by the average ranks of the acquired resistant cell line G12Ci values divided by their matched isogenic sensitive cell line G12Ci values. EMT module genes are color coded dark purple for clarity. Bottom right, GSEA plot for the pan-cancer EMT module. Although this was not a top 10 geneset, this was because of a large number of redundant cell cycle genesets that all scored very highly, pushing the EMT module down; however, it was still statistically significant as shown. (**G**) Spreadsheet screenshot of the TCGA BLCA dataset, with samples grouped by subtype and genes sorted by correlation to the basal module. Overall, 14 cancer modules are along the left, of which 8 show a clear basal gene signature (demarcated by the vertical black line). Core basal genes are color coded purple for clarity. Selected core basal module genes are shown at the right. (**H**) GSEA NES scores of a direct comparison of basal genesets in MSigDb c2 to our pan-cancer basal module (red).

**Figure 7 cancers-16-01886-f007:**
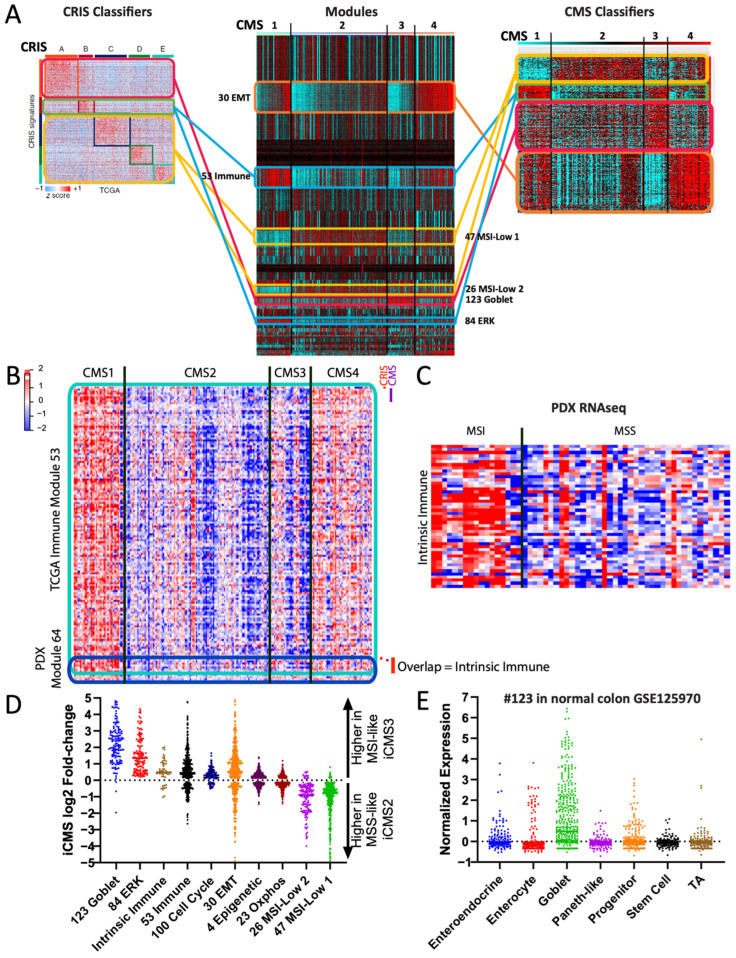
Modules can act as decoders between signatures. (**A**) COAD modules acting as a scaffold/decoder to compare the CRIS and CMS gene signatures. Lines are color coded to indicate which module each subset of genes are primarily members of (e.g., red for goblet). (**B**) Heatmap showing intersection of the TCGA-derived immune module #53 and the PDX-derived immune module #64, an overlap of 53 genes. (**C**) Heatmap of the same 53 overlapped immune genes (manuscript in preparation, D.M., S.K.), grouped by MSI/MSS status. (**D**) Fold-change expression of TCGA-derived modules (plus the 53 intrinsic immune genes) in scRNAseq-derived iCMS3 vs. iCMS2 subtypes. Each dot is one gene. (**E**) Normalized expression values of the TCGA-derived module #123 in scRNAseq data from normal colon epithelial cell types. Each dot is one gene. TA = transit-amplifying.

**Figure 8 cancers-16-01886-f008:**
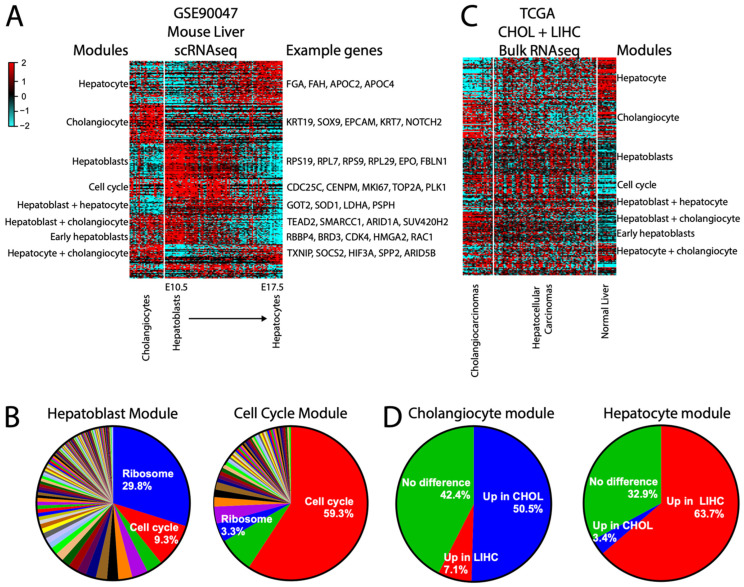
scRNAseq modules can inform bulk RNAseq data. (**A**) Heatmap of the largest modules derived from scRNAseq mouse normal liver data containing cholangiocytes and a timecourse from hepatoblasts to hepatocytes. Modules are labeled by the cell type they are expressed in. (**B**) Heatmap of the same modules applied to the TCGA CHOL (cholangiocarcinoma) and LIHC (hepatocellular carcinoma) combined datasets, including normal liver. (**C**) Pie charts showing TCGA-derived module memberships within each of the mouse scRNAseq-derived hepatoblasts and cell cycle modules. (**D**) Pie charts showing conservation of scRNAseq-derived cholangiocyte or hepatocyte module genes as up in CHOL or LIHC, respectively.

## Data Availability

All datasets used in this study are listed in Appendix A.

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
