# Peer review of "A Gold Standard-Derived Modular Barcoding Approach to Cancer Transcriptomics"

_cancers, 2024, doi:10.3390/cancers16101886_

Round 1

Reviewer 1 Report

Comments and Suggestions for Authors

The manuscript focuses on decoding transcriptome data by creating cancer-type-specific gene expression modules into flexible barcodes. Transcriptome data from TCGA and other sources were used to test the hypothesis.  The author has effectively justified the utilization of diverse statistical methods, and the results and discussion are adeptly presented. However, there is a need for greater clarification in the method section, giving rise to the following concerns:  

1) basis for choosing only 14 cancer types?

2) Line 627: What is top 5% per gene in "...gene correlations for any dataset, keeps only the top 5% per gene,...."?

3) The Methods section is not very clear. I will recommend dividing especially the "programs" part into specific subsections according to the workflow followed.

4) Did I miss sample information about the final TCGA RNAseq datasets that you used in the analysis? Did you use raw read counts or FPKM? Mention all these details. Did you use any preprocessing?

5) Line 82, please expand manual examination. How was that performed?

6) Line 186-221: In many places, Figure 2 is mentioned. Is it Figure 2 or Figure 3? Make sure.

7)  Figure 5, What is the significance of "*"? Improve the figure quality, too. 

Comments on the Quality of English Language

Minor editing, typo

Author Response

Response to Reviewer Comments

We thank the reviewers for their perceptive critiques, including their description of our work as “effectively justified”, “adeptly presented,” and “interesting.” Below we provide a point-by-point response to each comment, in blue.

Reviewer 1

1) basis for choosing only 14 cancer types?

We have now detailed in the Methods that the 14 cancer types were selected based on having the largest number of samples available (>300) at the time of initial analysis.

2) Line 627: What is top 5% per gene in "...gene correlations for any dataset, keeps only the top 5% per gene,...."?

We have now clarified the process in the Methods: the end result of the initial correlation calculation is a matrix in which each gene is correlated to every other gene. In order to identify the strongest correlation clusters, for each gene only the top 5% of genes with the highest correlation values to it are kept. All other genes for that gene are set to a correlation value of zero. This 5% cutoff prevents intermediately-correlating genes from “pulling” genes out of their “proper” module.   

3) The Methods section is not very clear. I will recommend dividing especially the "programs" part into specific subsections according to the workflow followed.

We have now provided an improved Methods section with better organization and detail according to the reviewer’s suggestions.

4) Did I miss sample information about the final TCGA RNAseq datasets that you used in the analysis? Did you use raw read counts or FPKM? Mention all these details. Did you use any preprocessing?

We thank the reviewer for this excellent observation. We have now clarified in the methods that the data we used are directly from the TCGA firehose portal (https://gdac.broadinstitute.org/), specifically the files that end in “illuminahiseq_rnaseqv2-RSEM_genes_normalized”, which provide TPM. We did not further preprocess the files, other than to remove the normal tissue samples, before correlation analyses.

5) Line 82, please expand manual examination. How was that performed?

Prior to module creation, our team had used MSigDb to annotate >1000 genes as being likely “core members” of known genesets. So, for example, genes such as TOP2A, MKI67, CCNB1, AURKA, MCM2, etc. were labeled as “cell cycle.” This annotation existed in a spreadsheet, which was then directly compared to our modules, leading to easily-visualized enrichments within many modules. In fact, our earlier annotation, which was not done in a systematic way, was one of the main motivations for creating the modules as way to derive annotations more rigorously and unbiasedly. This has been briefly summarized in the Results section.

6) Line 186-221: In many places, Figure 2 is mentioned. Is it Figure 2 or Figure 3? Make sure.

We thank the reviewer for catching these errors and have fixed them.

7)  Figure 5, What is the significance of "*"? Improve the figure quality, too. 

We appreciate the reviewer pointing out that “*as of 2020” was not sufficiently visible. To correct this, we have now also added the explanation to the figure legend. The figure quality has also been improved.

Reviewer 2 Report

Comments and Suggestions for Authors

In this research, authors identified gene modules that were detected in the TCGA datasets. Using the coexpression analysis, genes showing high correlations were defined as modules. For validation of utility of the modules, they performed various analyses. I think these results are interesting, but there are some issues to be discussed. 

1. The definition of the modules is too simple, and there was no benchmark analysis about defining the modules. In fact, there have been a lot of studies about gene expression modules, and their model should be compared with the current modules. 

2. Biological interpretation of the modules seem to be necessary. For application of the module information to the interpretation of the other datasets, more biological interpretations are required. 

3. It is hard to find the meaning of the "cancer module relationships". I think authors should provide novel findings or explanation using the identified cancer module relationhips. 

4. In the application of the modules to other datasets, authors should provide more objective measures for value of the modules. Although they added the explanations why the modules were useful in the analysis of the other datasets, they were too descriptive that are hard to decide whther the explanations were valid. Please presented the objective results (such as enrichment scores) for identification of the biological menaings of the modules when they are applied to the analysis of the other datasets.

Author Response

Response to Reviewer Comments

We thank the reviewers for their perceptive critiques, including their description of our work as “effectively justified”, “adeptly presented,” and “interesting.” Below we provide a point-by-point response to each comment, in blue.

Reviewer 2

  1. The definition of the modules is too simple, and there was no benchmark analysis about defining the modules. In fact, there have been a lot of studies about gene expression modules, and their model should be compared with the current modules. 

We thank the reviewer for the excellent suggestion of benchmark analyses such as overrepresentation analysis to improve the definition and qualification of our modules. After much thought, we have concluded that this question and the one in point #4 below are intimately connected and thus can be answered together. This is because using GSEA to benchmark our modules kills two birds with one stone: 1) it provides the “objective measures” requested in #4, and 2) the way in which we run GSEA directly compares our modules to the MSigDb genesets c2 and c5, allowing us to assess head-to-head performances of similar modules and genesets. Please see our response to #4 for further details.

  1. Biological interpretation of the modules seem to be necessary. For application of the module information to the interpretation of the other datasets, more biological interpretations are required. 

We thank the reviewer for their perspective, but we must emphasize that extensive biological interpretation of individual modules is not the goal of our study. In fact, we state in the Discussion that “it is not the goal of each module to represent “biological truths””, but rather to serve as useful mathematical constructs to point a user towards such truths.

This is especially important as – with the possible exception of the pan-cancer modules – many modules will have different biological interpretations in different contexts. For example, the scRNAseq hepatoblast module in Figure 8 makes sense in the context of embryonic liver but loses biological meaning in the bulk RNAseq liver cancer dataset, likely because no hepatoblasts are present. Yet nevertheless, other modules from the scRNAseq data prove useful in the comparison. In other words, the ultimate interpretation of these modules is best left to the end user. The modules that we have highlighted in the manuscript only serve as proofs of principle and proscribe a set of logical steps for other users to emulate to maximize the utility of our modules.

Indeed, we can sum up by saying that the goal of our study is not as much about the actual modules we have generated, but rather demonstrating that our overall modular barcoding approach is a novel and useful way in how to think about and characterize transcriptomic data. This is illustrated by our desire for users to download our module-generating programs to create their own modules. This is especially important over time, as single cell RNAseq data is becoming more readily accessible and higher quality, which will eventually lead to modules with higher and higher precision. It is our goal with this manuscript to set a blueprint in which modular barcoding can assist in creating, organizing, and utilizing future improved modules. We have further emphasized these points in the Abstract and Discussion.

  1. It is hard to find the meaning of the "cancer module relationships". I think authors should provide novel findings or explanation using the identified cancer module relationhips. 

We appreciate the reviewer asking for novel findings. However, we respectfully assert that we have provided many novel relationships with clear biological value. For example, the ability of our selected melanoma modules to clearly and consistently stratify patients by overall survival, which has not been shown previously (Fig. 4C); our identification of C16ORF59 as a high-confidence novel cell cycle gene (Fig. 5A,B); our identification of EMT as a high-confidence KRAS G12C inhibitor resistance mechanism that was not identified by the authors of the original paper (Fig. 6D); the identification of a novel basal module (Fig. 6E); and the identification of a novel tumor-intrinsic immune signature as inherently elevated in CMS1 subtype colorectal cancers (Fig. 7C, D). However, to further satisfy the reviewer, and due to their excellent suggestions, we now provide additional novel connections identified through GSEA analyses (see #4 below): 1) ribosome modules negatively correlate with immune checkpoint survival in the Liu melanoma dataset, previously unreported; and 2) a novel correlation in cholangiocarcinoma of the CC1 signature positively with epigenetic and oxphos modules and negatively with a basal module.

  1. In the application of the modules to other datasets, authors should provide more objective measures for value of the modules. Although they added the explanations why the modules were useful in the analysis of the other datasets, they were too descriptive that are hard to decide whther the explanations were valid. Please presented the objective results (such as enrichment scores) for identification of the biological menaings of the modules when they are applied to the analysis of the other datasets.

We thank the reviewer for this excellent idea. Following the reviewer’s suggestion, we have now conducted GSEA by inputting our modules into the existing MSigDb c2 (experimentally-derived) and c5 (knowledge-based) .gmt files, which allows us direct head-to-head comparisons with existing genesets (thus also addressing comment #1). We now provide the NES enrichment scores for each panel in Figures 6 (GSEA is only applicable to this figure, because GSEA requires a rank metric and the other figures are not based on rankings). We demonstrate that our modules frequently outperform or equal similar existing genesets.

First, in the Liu dataset in new Figure 6B, we show that both the melanoma-specific and pan-cancer immune modules notably outperform immune genesets in both c2 and c5, and furthermore that GSEA identified a high enrichment for both melanoma-specific and pan-cancer ribosome modules as negatively correlated with overall survival – a novel observation – and outperforming or equaling existing ribosome genesets. The melanoma-specific epigenetic module also scored highly as predicted by the colored modules in Fig. 6A.

Second, in the cholangiocarcinoma dataset in new Figure 6D, we show that our CCA1 signature module outperforms 2 existing CCA1 signatures in MSgiDb c2 (Oishi and Andersen), and that furthermore our modules identify a basal signature as enriched in CCA2 samples, previously undescribed and only brought to light by our modules.

Third, we validate the pan-cancer EMT module as significantly enriched in the resistant NSCLC cell lines. Although it is not one of the top-most enriched genesets, this is primarily because of the large number of cell cycle genesets in MSigDb that collectively outscore it – a general issue with MSigDb genesets any time cell cycle is highly significant. The GSEA plot and associated statistics are provided instead in new Figure 6F.

Fourth, we have added a new Supplemental Figure 6 focusing on the pan-cancer cell cycle module as a representative example of our modules’ strengths. In this figure, we directly compare the GSEA enrichment plots of the pan-cancer cell cycle to MSigDb c2 and c5 cell cycle-related genesets, using data from the resistant NSCLC dataset. We show that the genes in our module visually pack much tighter in enrichment than any of the knowledge-based c5 genesets and have much higher enrichment scores despite larger gene numbers than c2 genesets, together showing that our pan-cancer cell cycle module is more comprehensive than existing genesets while retaining very high accuracy, demonstrating higher overall performance.

Finally, in the derivation of the “pan-cancer” basal module in new Figure 6H, we compared our module to all existing basal genesets (all derived from breast cancer) across BRCA, BLCA, LUSC, and PRAD. Although our module did not always score the highest, it scored the most consistently across cancer types, suggesting that our basal module is indeed more applicably pan-cancer than the breast cancer-derived basal genesets.

Note to editors: Figure 6 is now expanded so that it fills 2 pages, and thus we have split it into Fig. 6 and 6b.

Round 2

Reviewer 1 Report

Comments and Suggestions for Authors

All of my comments were addressed.

Comments on the Quality of English Language

Minor editing is required.